# Systemic Design for Policy-Making: Towards the Next Circular Regions

**Carolina Giraldo Nohra \***, **Amina Pereno** and **Silvia Barbero**

Department of Architecture and Design, Politecnico di Torino, Viale Pier Andrea Mattioli, 39, 10125 Torino, Italy; amina.pereno@polito.it (A.P.); silvia.barbero@polito.it (S.B.)
\* Correspondence: carolina.giraldo@polito.it

**Abstract:** The vast transformation the circular economy that will occur in the upcoming years inevitably will change the EU panorama, designing new scenarios from an economical-social-environmental perspective. To best build a circular economy, it is necessary innovative policy-planning with a holistic and systemic perspective that fosters a cohesive and smooth transition to circular business models. This paper explores the impacts of circular economy policy design processes driven by a systemic design and how this expertise could ease innovative and effective paths for policy-planning on a circular transition in EU regions. This examination of systemic design features recent approaches to design as a discipline addressing complex problems, and the literature review on systems and design thinking for sustainable development, and policy design, focusing on existing barriers to circular economy. The discussion is narrowed to the specific case study in which the systemic design methodology is applied to provide a path for five European regions towards the CE: the Interreg Europe RETRACE (A Systemic Approach for Regions Transitioning towards a Circular Economy) project. Including an in-depth examination of how systemic design can address current barriers for a circular transition within an effect in the short, medium, and long-term policy horizon in the transition of the European regions towards the circular economy.

**Keywords:** design for sustainability; systemic design; systems thinking; policy design; circular economy

## 1. Introduction

Today—through times of frenetic change—the world is experiencing growth of interconnected megatrends. This increasing discussion of a state of complexity, described in reports such as PWC (PriceWaterhouseCoopers) [1] includes radical transitions in economic power, accelerated urbanization, climate crisis, resource scarcity, technological breakthroughs, and erratic social transformations. The interconnected nature of such critical drivers has a significant impact on how governance operates, requiring more future-oriented and sustainable-oriented policy actions. These megatrends are embedded as the so called "wicked problems", which—by their nature—cannot be solved as they are because of their consistency and worsening [2]. Hence, to tackle this wickedness, sustainable development urges the balance of ecology, economics, politics and culture dimensions [3]. In order to achieve that proportion is required a holistic frame of reference [4] which to ensures intergenerational and intragenerational fairness [5]. Today this approach to sustainability resembles to be incarnated by the circular economy (CE) concept in opposition to the current inefficient linear economy [6].

The CE fosters the transition from a linear model towards a more sustainable economy, focusing on the benefit for the environment and the society [7]. This concept has an extensive background of scientific fields and theories that have surrounded it from the begging [8–15], in particular the concept of "eco-effectiveness", conceived by cradle-to-cradle [16] and the industrial ecology [17]. Considering

this scenario, the definition of the CE concept provided by Korhonen et al. [18] brought a complete perspective of this topic related to sustainable development and sustainability science:

> *"Circular economy is an economy constructed from societal production-consumption systems that maximize the service produced from the linear nature-society-nature material and energy throughput flow. This is done by using cyclical materials flows, renewable energy sources, and cascading type energy flows. Successful CE contributes to all the three dimensions of sustainable development. CE limits the throughput flow to a level that nature tolerates and utilizes ecosystem cycles in economic cycles by respecting their natural reproduction rates".*

Such a definition underlines how the wickedness of the linear economy has raised unpredictable risks as resource prices, job instability, and supply disruptions. Considering the complex and interconnected nature of those challenges, the road towards a CE requires a systemic perspective, where the depth understanding of the complexity comes from the number of variables and relations created in the environment. For this reason, there is an increasing recognition of the urgency to propose effective policy strategies to transition towards a CE creating new socio-technical systems [19].

In the case of the European Union (EU) and its current challenges, this transition towards CE results particularly appealing. As, it is a chance to foster competitive advantages on a sustainable basis with the potential to create an income benefit of EUR 1.8 trillion by 2030 [7] and over one million new jobs across at EU level by 2030. Applying CE principles across all sectors and industries within a foresight perspective could imply a decrease in the environmental, social and economic pressures, increasing the EU regions strategic autonomy [20,21].

The European Commission has targeted the CE as one of the main objectives through the settled goals on sustainable development, low carbon targets, and resource efficiency [22]. To that end, on 2015 it released the circular economy Package, whose aims is through the integration of policy proposals on waste management, landfills, and recycling and reuse, encouraging a shift to a sustainable economic paradigm in the EU. On an EU scenario, it is clear the transition is underway on a vast scale with transforming economies, governments and societies in complex, interconnected and unpredictable ways. Nevertheless, the execution of this potential is hampered by various barriers such as economic (instability of the market), social (lack of knowledge to distinguish opportunities) and regulatory (policies that inhibit the reuse waste) [7]. To accomplish CE goals at the EU level, it is imperative to overcome these barriers at a governance level, starting from a deep comprehension of the system complexity and the ecology of its relationship [23].

Today, policymakers are in a crucial moment on delivering effective CE policies that should be synchronized with the requirements of each context on relation to major adjustments in production and consumer behavior [24]. These policy strategies should be based on an innovative model of governance open enough to new structures of rules and actors capable of combining top-down and bottom-up processes. In addition, capable of activating new mechanisms of decision making, such as design thinking, participatory, and systemic approach towards a CE.

The transition towards a CE must imply active cooperation between the different stakeholders [25], as well as the integration of various policy interventions that will not imply "silver bullet" solutions [7] or classical work frames of a single organization [26]. This policy-making process must require a mediator that encourages people to "think outside the box" and generate such disruption [27]. In those circumstances, is where the role of design can be crucial, due to its nature to deal with complex scenarios [28]. Designers as mediators have the skillset to anticipate future situations and generate innovative outcomes promoting new approaches to complexity. The implementation of several design methods and thinking in the field of policy-planning are evident in many practices of the last decade and the foundation of various policy labs, run by designers [29].

To arrive at this level of decision-maker, the designer has evolved its role, providing skills and capabilities for sustainable development and managing the complexity [30]. Over the last decade, the design discipline has focused more on sustainability as a system of resilient relationships instead of a

characteristic of individual components in systems [31]. In this path, the systemic design (SD) discipline has emerged as crucial expertise which provides practical tools to approach complex scenarios with a holistic perspective, while supporting active cooperation among involved stakeholders. The systemic designer uses the traditional design skills, such as research methods, process thinking, and visualization practice, with the implementation of new advanced skills, such as mapping schemes for the complex mediation between different knowledge, divergent thinking, co-design processes, critical thinking and creative making [32]. From this perspective the SD is gaining momentum in several fields, especially in the policy-making area. Although designers are increasingly involved in policy design processes at EU and international level, there is no evidence of how they can boost bottom-up processes aimed at a radical shift of local policies, as the CE requires.

Therefore, the article addresses this research gap exploring how the SD can support the creation of policy actions rooted in the regional context but contributing to an interregional transition to the CE. More specifically, the article poses two research questions:

- RQ1: What are the main barriers to the transition to the CE and what challenges do they pose to the disciplines involved in this transition process?
- RQ2: How can systemic design support this transition? Through what tools can it help to overcome the barriers to the CE and which results can systemic design bring?

To answer these questions, the research carried out a desk research phase to investigate the issues raised by RQ1 and a field research phase to address the challenges posed by RQ2. The topic addressed is extremely complex and the shift from the design to the implementation phase is challenging. However, the article explores the theoretical contribution of SD methods in the CE field, proposing a validation of these methods through a four-year interregional project that, although not exhaustive, provides a fundamental contribution to the state of the art and the literature in this field.

To better illustrate this research path, the article is structured in four sections: Section 2 provides an overview of the methodological path followed; Section Section 3 presents the results of desk research and especially the role of SD methodologies in addressing the transition of complex systems, focusing on the CE and the emerging barriers to it; Section 4 presents the results of field research delivering a broader examination of a pilot research project that applied SD to regional policy design, involving five EU regions; Section Section 5 discusses the results presented and explores the impacts of CE policy design processes driven by SD and how this expertise could ease a bottom-up transition to CE.

## 2. Materials and Methods

This article aims to assess a better comprehension of the SD discipline as supportive expertise to unveil innovative paths for policy-planning towards a CE in Europe, through the discussion of the results of the RETRACE Interreg Europe project. In order to do so, the research method applied a two-stage approach: a literature review to define by inductive way the systemic methodologies applied to policy-making and the main barriers to CE, an analysis of RETRACE case study and outcome impacts to evaluate the effectiveness of the methods applied and the triggering and hindering factors to policy-making towards CE. This process makes it possible to identify the most suitable methodological tools to promote a systemic approach to the CE, through a dual inductive-deductive method (Figure 1).

The literature review introduced with an inductive analysis of scientific contributions on the design discipline evolution approach for wicked problems and sustainable development featuring recent approaches of systems thinking to design as a discipline addressing complex problems. Furthermore, it comprehends the main targets of design today on delivering sustainable services and systems, thereby it provides a detailed of the SD discipline and methods. Afterwards, the analysis continues towards the influence of design methods on the policy-making field, to comprehend the advantages of how approaches such as the SD can serve Policy design processes, and which are the most suitable in relation to CE. Therefore, to prove the suitability of the SD as a methodology to address the CE policy field, a brief literature review on the common barriers to transition to a CE was executed. The examination

was followed by an analysis of how each methodological step of SD addressed the main CE barriers identified in the literature.

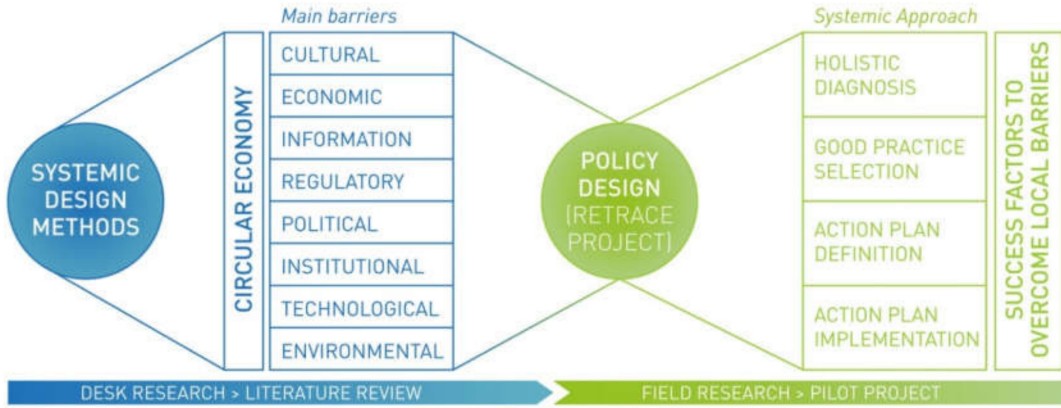

**Figure 1.** Graphic representation of the research methodology.

In the second stage of the examination, the argument is narrowed by the Research Exploration: RETRACE Interreg Europe Project (2016–2020) case study in which the SD is applied to provide effective policies for regions towards the CE. For this research, the case study presents a general project overview on regards partners, stakeholders and timeframes. Afterwards, it explained in detail the SD approach on the project framework including a focusing on the methodology for key stages:

- Holistic diagnosis (SD context analysis method based on qualitative-quantitative data sets);
- Good practice selection (field visits to collect CE good practices and identify policy gaps);
- Regional action plans (policy actions to foster CE at a regional level).

The goal of presenting that project framework is to narrow the discussion to the policy impacts and outcome on RETRACE partner regions and determine a basis to compare and relate. The discussion of such impacts can illustrate the spectrum of challenges that the SD policy-making process can face in different scenarios underlying the barriers faced in process implementation.

To better comprehend such outcomes within a vision on multiple time horizons, this paper delves into an overview of the a short, medium, and long-term policy time frame in the transition of the EU regions towards CE. The SD provides a transdisciplinary scenario that brings together governments, industry, and citizenship, in order to face the contemporary environmental and social challenges that such approaches had face in practice.

## 3. Systemic Design for Circular Policy-Making

### 3.1. Wicked Problems and System Transitions

Today the conversations around design discipline are more than ever linked to critical areas of the sustainability field and practice. Such dialogs in innovation are considered the main leverages for change in society, industry, local communities, and government [33]. In this view, the design discipline has trespassed its frontiers. Whereas historically, this discipline was mainly related to the development of the material culture and the creation of objects, starting from the lasts decades design has broadened its field to services and systems. This process portraits the evolution of the design for sustainability, starting from product innovation to product-service system innovation to spati-social innovation and socio-technical system innovation [31].

In recent years, the number of designers that have developed skills and methods towards intangible targets like sustainable behaviors and development has dramatically increased. This trend on the discipline indicates how design is transitioning into dematerialization and getting near to solutions for wicked problems on complex social, environmental, and even political problems [34].

Such transformation mindset occurs because designers have recognized their impact and responsibility on the globalization effects regarding mass production and consumption. The cross-cutting approach between design for sustainability has gained momentum, delivering new approaches to the role of the design in our society. On this transition, design discipline has become more participatory as it has widened its field of practice from a single user to communities as a user and even governments, delivering sustainable services and systems [35,36]. This transition process of design discipline towards services and systems has been described by Buchanan (1992) [37] who focused on the role of design in approaching the wicked problems, emphasizing the ultimate goal to transition to a resilient, fair, and sustainable society. Therefore, it is vital to understand the wickedness in governance that design discipline is facing today. Consequently, an extensive literature has defined the wickedness of these issues inside the complex systems as [38]:

- Nonlinear, which means that the whole is greater than the sum of its parts;
- Adaptive, referred to the resilience of the system and its parts accommodate overtime to the transformations inside the context;
- Self-organizing, the parts of a system can rearrange autonomously;
- Emergent, which means the difficulty to anticipate the system output from actions carried out at the component level.

Such comprehension of wicked problems allows an overview of a predominant complex scenario which comes from a process-based, multi-scale, and systemic approach [39]. Furthermore, enables a more in-depth critical analysis for profound leverage for change from a social, cultural, and organizational perspective [40], which could be the key to deliver effective policies.

*3.2. Systemic Design Methods for Implementing Resilient Systems*

How has design discipline built a systemic approach that addresses today? This relationship has been influenced drastically in the last decades, the last within the integration of the general systems theory [41] and the system thinking into the design discipline. Therefore, the designer conceives the problem-solving process within a systemic thinking perspective, which means looking at the whole, analyzing single components and focusing on relationships. In practical terms, it requires to zoom out from an individual part and examining that part's connection to its surroundings and other ecosystems. From the implementation of Systems Thinking into the design discipline, has emerged the SD, which focuses on a holistic approach applied to projects for artefacts and services, allowing a broader view of scenarios and their wicked problems [42]. The SD blends the designer skills before-mentioned as research, problem-solving methods, and visualization practices and creates innovative reconfigurations for complex services and systems. The "Handbook of design for sustainability" [43] underlines this concept presenting the need for these approaches to wicked scenarios, recognizing how the design for sustainability can only achieve within the innovation at a system level, including multiple stakeholders like communities, governments, and industries.

Although SD has been applied to different socio-technological sectors, its implementation in the development of sustainable territorial systems enabled the creation of specific tools aimed at holistic analysis and design. Furthermore, has allowed to envision and design the flow of material and energy from one element of the system to another, transforming outputs of one process into input for another one to obtain zero emissions and generating resilient systems [44]. This approach supports active cooperation between local stakeholders that results in new, locally-based, value chains [45]. In this perspective, it has created a synergic link between the productive and natural context of the given territory, at the same time reinforcing the socioeconomic systems connected to that territory, within a long-term vision.

From this perspective, the SD acknowledges contexts to be understood in a deeper and broader perspective, surpassing the challenge of the small scale through a holistic approach. This overview considers in parallel the small and large scale into specific leverage for change while taking into

consideration the bigger picture. The SD within a territory aims the generation of new relations among the components enabling the visualization of the hidden potentialities: as a result, SD boosts a proactive collaboration among local actors and enhance future production activities [44]. Such outcomes lead to a better comprehension of the capabilities of the discipline to foster new sustainable and resilient economic models. The SD methodological approach is based on six main steps [46]:

1.  Holistic diagnosis: the first phase is based on a combination of desk and field research, which aims to investigate the given scenario from an economic, social and environmental aspect, considering the flow of energy and matter;
2.  Definition of problems and leverages for change: begging with the framework defined in the holistic diagnosis, connections and impacts are analyzed in order to identify potentialities and criticalities of future scenarios. The challenges are approached as leverages for change from which the system can be defined and initiated;
3.  Design the system: a new production model is designed whose aims is to tend to zero emissions by optimizing energy and material flows and by valorizing the waste as resources;
4.  Outcomes Evaluation: the assessment of the environmental, economic and social benefits belonging from the new production model;
5.  Implementation: apply the proposed designed system in the given context and consequent estimation of the new economic activities feasibility;
6.  Results analysis and feedback: assessment of the implemented system and ensuring to be autopoietic.

Systemic designers approach this process by gathering qualitative and quantitative data to be able to map the context within a multidisciplinary perspective [47]. Thereby they utilize graphic visualization tools to display the interconnections of the scenario components, rising number of interpretations for that context. The aim is to generate a broader range of probabilities to create innovative solutions and problem-solving. From this perspective, designers can address various scales to visualize challenges and conceive new strategies. Finally, they narrow a concrete solution through the following abilities [48]:

*   make information accessible and straightforward, by managing a significant amount of information and making them available to an end-user by easy-reading diagrams, schemas, and scenarios. (i.e., IDEO cards, giga-maps, etc.);
*   think creatively, and if used to policy-making, could lead to innovative strategies;

Create connections in complex systems, to allow a broader examination of the challenges and to support a transdisciplinary approach. The designer can interconnect the components of a system to create innovative business ventures, products, and services.

In this scenario, it is clear the role of designers as mediators since they enable facilitating and mediating different competencies [28]. The active role of a designer in the process of policy-making unlocks a capacity of organizational versatility and anticipation. In this view, the human-centeredness is the main component that Design includes the process. Hence, designers also underline the importance of the aforementioned qualitative features as culture, uses, or local resources.

### 3.3. Systemic Design and Policy Design for Governance

So how the SD can address the wickedness on governance? As society evolves faster today, governments struggle more and more to adapt to the erratic global changes. This situation is prevailing over time as governments traditionally tend to approach these problems in "silos" and, instead of fostering preventive policy systems, they often opt for reactive measures. Hence, the current policies which are aimed to regulate such systems on different levels (local, regional, national, and international) are missing a wide range of factors [49]. Therefore, the current focus of policies is not the most efficient since it entails a top-down approach that does not consider the final users, which are the citizens,

even though the main purpose should be *"Policymaking is the process by which governments translate their political vision into programs and actions to deliver "outcomes"—desired changes in the real world"* [50]. On this path design discipline, is more embedded in the everyday complexity, a system with infinite relationships that connect people, companies, and governments. A scenario in which designers can intervene with their skillset and problem-solving approaches to assess the wicked problems of the everyday system encompassing a variety of alternative frames [51]. As a result of this approach on wicked scenarios, the solutions are narrow into strategies by policy design.

To approach the "desired changes," it is essential to regard both approaches: design and policy-making as a problem-solving process. Consequently, is necessary to implement new strategies like the ones on policy design which are based on different design methods from design thinking, co-design and SD, which provide a different overview of understanding policy problems that address long-term vision and strategy [52]. Furthermore, facilitating the needed system overview also implies a network of multiple relations on which policy-planning aimed to enable such issues: "Systemic and interconnected problems need systemic and interconnected solutions" [53]. At the same time, the policy design promotes a comprehensive combination of research methods from different disciplines like data science, anthropology, and systems thinking that generate multidisciplinary synergies [29]. More importantly, this kind of policy-planning method considers vital that bottom-up and top-down approaches must find ways to coexist with a common objective [54]. In this perspective, it considers all the actors involved in the policy-making process should be able to communicate and work together with the same goal.

A tangible demonstration of this processes has taken place on the so-called "policy labs" (i.e., Public Policy Lab in New York, EU Policy Lab in Brussels, The Policy Lab UK Cabinet Office). They are commonly configured inside government administrations and adopt facilitation strategies through co-design techniques or collaboration toolkits (i.e., IDEO cards). Consequently, it makes policy tangible for all stakeholders in a decision-making process. At the same time, it promotes a horizontal dialog among all stakeholders, generating innovative and effective decision-making for policy design approaches. This engagement of stakeholders supports an exchange of strategic thinking process that leads to the definition and implementation of adequate policy-planning. To achieve these participatory processes and design effective policy strategies, the SD involves other methodologies such as: design thinking, co-design, user-centered design, and participatory design [42]. These approaches have in common the active engagement of users, generating an innovative decision-making process which turns the end-user into the focus of the policy-making formulation system [55].

Nevertheless, it is true that on the current Policy design field, the application of SD still has a lot to explore as an emergent discipline. The involvement of designers in policy-making today faces different the current linear governance system, by leaving space to the openness and cooperative methods of design, creating resistance from civil servants, policymakers and institutions. Moreover, the opposition of external views, into governance topics which have traditionally developed "behind closed doors". Lastly, the demand for immediate solutions that differs from the long-term view that an innovative action requires [55,56].

### 3.4. Addressing Circular Economy Barriers with a Systemic Approach

In this view, in this era of the climate crisis and a linear economic system, the need for SD and the potential of innovation policy addressing wicked problems has never been greater. In the last decades, the attention to "wicked problems" has increased in the field of policy research [2]. This growing interest also comes as a consequence of the increasing interaction between design and policy-making, which is a fast-evolving phenomenon at an international level [29]. In addition, from a methodological point of view, the SD aim is increasingly understood as practical and necessary within the policy design for the CE, since government bodies are aware of the need to adopt anticipatory actions on governance that take policy-planning one step forward to achieve sustainable regional development.

In this perspective, it is essential to underline the wicked problems or main barriers to transition to a CE. In order to understand how the SD tackles or approaches such aspects that can hamper CE policy-making process. Different sources have highlighted the wide range of the obstacles the CE has [57–61]. However, Williams [62] and Vermunt [63] delivered deliver a more holistic perspective of the problematics that a transition to a CE will imply. In their analysis, both examine how context influences the difficulties of implementation. Mainly, they go beyond the technical aspects, but they argue more profoundly about the cultural and societal aspects challenges and how these alter with context. Within the scope of this research, it was combined the vision of CE barriers from both authors in the following Table 1.

**Table 1.** Main barriers to circular economy implementation.

| Type of Barriers | Description of Barriers |
| --- | --- |
| Cultural | Current value and norms, current social practices, cultural diversity, public unawareness of resource cycle, public unawareness with natural environment, current lifestyles |
| Economic | Economic viability, prospective resource value is uncertainty, need of financial incentive, financial risk, global supply chain, cost of dealing with pollution, high land value and isolation of low-value activities, restricted demand for looped sources, health and safety risks, low price of raw material, absence of public expenditure and dependence on private expense. |
| Information | Deficiency data availability, lack quality of data, deficiency of information, loss of trust in information transfer and collect. |
| Regulatory | Absence of supportive framework, emerging models for looped resources, need of multilevel regulatory framework, |
| Political | Neoliberalism, require for long-term political support, clashing priorities, absence of combined approach to policy-making. |
| Institutional | Fragmented government, cultural and structural inertia, absence of cross-sector alliance, Separate performance of services, private actor appointment, absence of institutional capability, managing authorities with limited controls/capabilities/resources, absence of autonomy amongst local stakeholder, absence of commitment with civil society, absence of trust in policymakers |
| Technological | Absence of dissemination on circular planning and design methods, technical limitations, absence of operational conditions, modelling resource flows, current linear resource flows |
| Environmental | Pollution of environment, long-period to renew ecosystems, depraved urban resources |

With this in mind, it is possible to comprehend what potential levers must be regarded when promoting a transition to a CE in a territorial system. Therefore, the SD is understood as a method that can comprehend more holistically the implications that CE has in a given scenario. For that aim, the SD can identify and comprehend the CE barriers and levers from a systems-thinking point of view. Afterwards, it unleashes the potential strategies that can promote an autopoietic and resilient system. So, to comprehend better how SD method addresses the CE barriers, Table 2 explains how each SD step addresses specific CE-related challenges.

This analysis brings an overview of how the SD offers a practical approach to the policy-making process on CE. Lastly, it is to underline the key role of the systemic designers since they were the experts that provided the methodology. As providers and mediators, their flexible roles allow a Quadruple Helix approach where university, industry, government, and civil society cooperate to co-develop strategic decision-making process. In this regard, the following exemplification on the RETRACE Interreg Europe project will expose how the advantages and challenges of CE Policy design processes can bedriven by SD.

**Table 2.** Systemic design methods related to circular economy challenges.

| Systemic Design Method Steps | CE Challenges | CE Challenges Addressed by Systemic Design |
|---|---|---|
| Holistic diagnosis | Mapping complex cross-sectoral systems to provide information for decision-making | This analysis aims to deliver a complete system perspective of the scenario, from the quantitative to the qualitative perspective. Delivering a complete overview of data and the total amount of resources and stakeholders the system implies to address a cohesive circular model. |
| Definition of problems and leverages for change | Lack of accuracy on addressing all the relevant assets/sectors to transition to CE | After approaching a system perspective, the definition of challenges and leverages will be more accurate because the future CE will be addressing and exploiting aspects tailored to the territory. |
| Design the system | Challenge to design innovative and sustainable socio-technical systems | A new system perspective is designed. Having in mind CE strategies that will valorize all territorial resources. Promoting a systemic transition to CE. |
| Outcomes evaluation | Need of a long-term vision of multi-governance policy strategies | An assessment of the environmental, economic and social benefits, implies a foresight vision on the CE strategies. |
| Implementation | Incapability to execute sustainable innovation actions in a long-term | Apply the proposed system in the given context, and this will imply the implementation of short- and long-term CE strategies. The purpose of this is a gradual transition to a CE system that will deliver more realistic and consequent feasibility of the new economic activities. |
| Results analysis and feedback | The need of horizontal dialogs between the quadruple helix components | The assessment of the implemented system and ensuring autopoietic is the ultimate proof to know a territory has transitioned to a CE. |

## 4. Research Exploration: RETRACE Interreg Europe Project

The SD expertise is rising as relevant to policy design fostering better governance towards CE [64]. An exemplification of this is the RETRACE project (2016–2020) (A Systemic Approach for Transition towards a circular economy) funded by Interreg Europe ETC Programme, under the 4.2 specific objective—improving resource efficient economy policies. This funding program aims at improving the implementation of regional development programs and policies by promoting experience exchange and policy learning among different regional actors. RETRACE aims at promoting SD as a method for local and regional policies to move towards a CE, according to which waste from one productive process becomes input into another, preventing waste to be released into the environment. The goal was to develop a regional action plan (RAP) for each of the regions involved, standing the research on contextual data and matching them with the good practices experimented in the other countries.

The coordination of the project comes from the synchronized work between universities, Development Agencies, government offices, associations and public companies. Their main focus is to undertake the EU priority of transitioning towards a CE keeping the subject set up by the "flagship initiative for a resource-efficient Europe" for a change to a resource-efficient, low-carbon economy to accomplish sustainable growth. The project lead by the Department of Architecture and Design at the Politecnico di Torino, involves eight private and public partners and more than 70 stakeholders from Piedmont (IT), Basque Country (ES), Nouvelle Aquitaine (FR), Northeast Romania (RO) and Slovenia (SI) to foster the cooperation between regions of EU countries.

### 4.1. Stakeholder Engagement Process

The involvement of stakeholders was essential to ensure the participatory development of new policy actions, the effective implementation of these actions, the monitoring of RAPs with regular and constant feedback, the dissemination of results and the creation of a supportive background for the transition to the CE. For this reason, the stakeholder engagement process consisted of three phases:

1. **Stakeholders mapping and selection.** The partners of each region have carried out a stakeholder mapping to identify the stakeholders that can influence the transition to the CE in public administration, business, research and education, and civil society. The selection was based on a mutuality perspective, identifying the reasons why each stakeholder was important for the project and, at the same time, the benefits stakeholders could gain from collaborating in the

project activities. This enabled gaining awareness of the stakeholders' points of interest and ensuring an active and constant involvement during the four-year project.

2.  **Creation of a regional Stakeholder Group**. The second phase focused on the creation of a Stakeholder Group in each of the five regions involved. By joining the group, the stakeholder committed to taking part in the periodic meetings of the regional working group, where research activities and RAPs were discussed.

3.  **Stakeholders involvement in research activities.** The third phase was the engagement of the Stakeholder Groups in the project activities, especially in the interregional exchange of experience. Stakeholder participation in international Field Visits and dissemination events was voluntary, so that people actually interested in acquiring new knowledge on CE could be involved. However, their experience was shared within the Stakeholder Group, so that personal narratives could spark the debate on transversal issues and good practices transferable to different regions. This brought positive results in terms of capacity-building in support of regional strategies towards the CE.

Figure 2 shows the overall composition of the stakeholders involved in RETRACE: the greater presence of company representatives is linked to the heterogeneous nature of this component, which required a large number of participants representing different industrial sectors. On the whole, the selection of stakeholders enabled fully understanding the quadruple helix model of each region. Stakeholders' engagement was critical to develop proposals grounded in the local context. Moreover, the multi-stakeholder dialog can support a bottom-up process for designing effective transition policies.

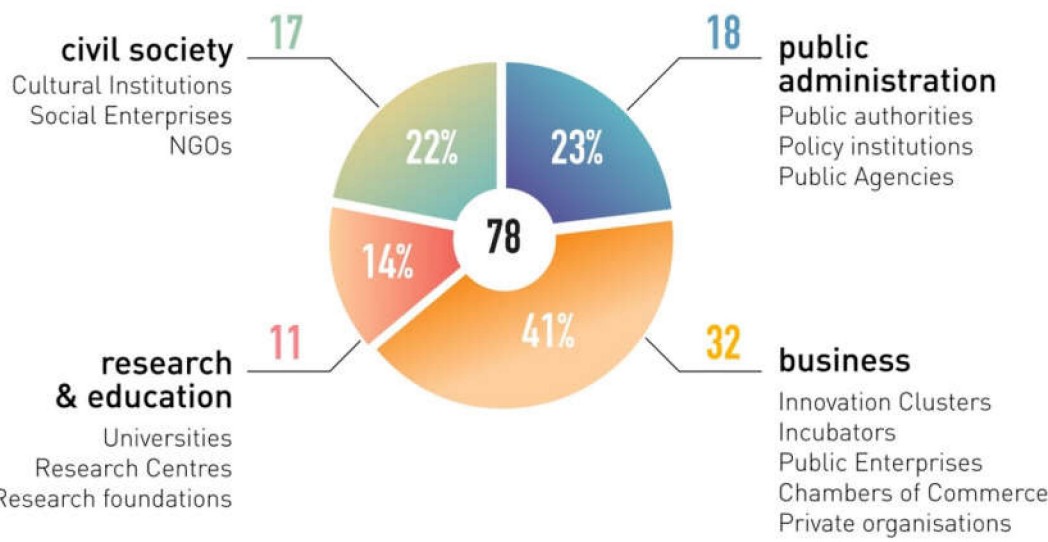

**Figure 2.** Composition of the stakeholder groups involved in the project.

*4.2. Systemic Design Research Methodology for Circular Policies*

The timeframe of RETRACE (2016–2020) is characterized by research (Phase 1) and implementation (Phase 2) (Figure 3). Phase 1 took place from 2016 until 2018 with a series of intense research activities: holistic diagnosis, good practices selection, and elaboration of the RAPs. Phase 2 took place from 2018 to 2020 and focused on implementing and monitoring the RAPs, assessing the main results and outcomes of the policy actions developed in phase 1.

Phase 1 adopted methodological tools aimed at obtaining a systemic vision of the state of the art in each of the five regions involved, enabling to understand the strengths and weaknesses of each area (holistic diagnosis) and to highlight the business and policy practices currently in place to promote the CE (GP selection). The systemic methodology adopted triggered a policy-making process to develop tailor-made strategies able to meet the needs of each region, while pursuing a common goal

of transition to the CE. Below, the three methodological tools are explained to understand in detail how the SD laid the foundations for the policy design process.

**Figure 3.** Project development: RETRACE Interreg Europe project timeline and phases.

### 4.2.1. Holistic Diagnosis

The focus for holistic diagnosis was to deliver a better comprehension of the main complex industrial systems (productive chains) of each region. This holistic panorama facilitates the policy-making processes on identifying the opportunities that can foster a transition into CE, where waste (outputs) from one productive chain part become the inputs for another one.

Holistic diagnosis is at the forefront of the SD method and enabled the collection of qualitative and quantitative data, to afterwards execute an analysis of interactions between all the components [48]. This tool aims to enhance the capabilities of policymakers to approach public issues, through a qualitatively innovative driver in the process of policy-planning. It is a supporting instrument to promote public sector effectiveness and innovation. Nevertheless, as part of the implementation of the SD in policy-making processes has proven the key role of these approaches on participatory processes.

It is important to underline that the SD methodology was shared and put in practice by all the partners of RETRACE, generating new mechanisms of sharing knowledge and experiences, on the local and interregional scale. On the RETRACE scope of the holistic diagnosis was to emphasize the critical aspects, as well as hidden potentialities, through a three steps process:

1. **Regional framework:** based on the data collection from quantitative data of official databases (e.g., Eurostat) about the territory characteristics and the industrial sectors, we also included qualitative data found on reports and on-site interviews with local stakeholders. As a result, the data collection presented through infographics, schemas, and gigamaps [65], which enabled easier fruition among a multidisciplinary team (Figure 4). This visualization delivered by the systemic designers, perform a vital role in the process since complex data (wicked problems) are made accessible and usable through a visual simple and common language.

2. **Analysis of current policies:** this overview of the policy state-of-the-art addressed traditional sectors on environmental sustainability in the areas of waste management, energy, and environment. The purpose was to underline the potential policy gaps that could inhibit the transition towards CE. In addition, each region regarded their current Smart Specialization Strategies (EU N° 1303/2013 of the European Parliament and of the Council, national or regional research and innovation (R&I) strategic policy framework) and development goals, among them a low-carbon CE.

3. **Analysis of the principal economic and industrial sectors:** the overlapping of the previous steps the policy instruments analyzed and the context information, deliver potential synergies at a systemic level between different value chains or processes at a regional and interregional scale. This action enabled the peer reviews discussions around the regional data presented and generating a knowledge transfer to the other partner countries.

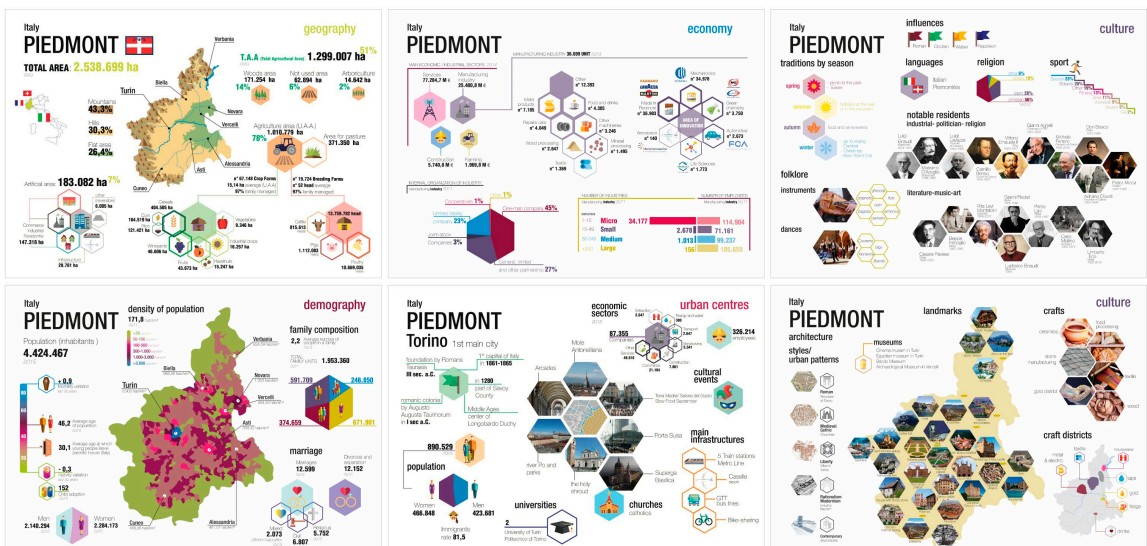

**Figure 4.** An example of holistic diagnosis infographics [48].

### 4.2.2. Good Practice Selection

The field visits across the partner regions focused on good practices related to policy that supported CE development rather than more traditional projects related to the sustainability area. At the field visits, partners and selected stakeholders of each region discussed the good practices analyzing their strengths and weaknesses to transfer this knowledge to the partner countries and understand how to solve the identified policy gaps. The selected good practices were linked to foster local resources understood as territorial potentialities. A total of 65 good practices of CE and SD were exchanged, out of which partner peer reviews selected 30 good practices. These good practices were chosen based on the degree of transferability and inspiring perspective [66].

### 4.2.3. Regional Action Plans

The RAP in CE for each partner region was one of the main milestones of the project [64]. These strategies described the regional priorities addressing a range of CE policy gaps in the five regions. In addition, it includes planned measures and support actions for the implementation period (2018–2020) and beyond. These future-oriented vision actions are pointless if not followed by a real implementation. The success depended on the path that led to the definition of RAPs that was defined by the SD.

The holistic diagnosis assessment of the regional context on regards the CE was essential to elaborate a RAP for each region. After matching the results of the holistic diagnosis and the experiences brought from the good practices, lead to the identification of six policy gaps common to the partner regions which the RAP approached. These explore the fields of intervention that could be tackled to support a CE and focus on the several aspects that imply a transition to a CE in such regions [66]. The identified policy gaps were the following:

1. **Support collaboration between sectors**: this policy gap regards the eligibility rules for the open industrial calls, which sometimes bans certain actors from different sectors to take part—hampering the creation of local value chains from the output–input principle and the technology transfer between regional stakeholders;

2. **Raise and knowledge of operators concerning CE:** the limited amount of actions for operators to raise their participation, awareness, and knowledge on CE. This is highlighted as a key issue as it compromises the development and success of CE related strategies;

3. **Policy regulations on CE:** there are unclear and disharmonized policy regulations on CE, particular on by-products production conditions at regional, national and EU levels;

4.  **Tailored policy measures on CE:** inside the EU funding schemes the CE is a transversal topic, nevertheless, it is needed to generate tailor-made policy measures and calls that address CE directly;

5.  **Policy in support to business and market development for CE activities:** apart from developing CE projects, it is necessary to foster the generation of CE businesses models, in order to encourage the market to the reuse of by-products. These strategies are essential to address the waste management issues towards the success of the CE.

6.  **Policy focused on Small and Medium Enterprises (SMEs) and micromanufacturing:** This gap focused on two main challenges, i.e., the scarce tailored support on CE for SMEs transition and the limited assistance for the creation of micromanufacturing processes sized to the local context.

The RAP was a strategic document in which these policy gaps have been approached through different actions (see paragraph 4.3). To understand their impact, and measure the effectiveness of each action, specific indicators were identified.

Considering these policy gaps the RAP actions recognize that a transition to a CE is not possible without fundamental changes in areas such as: multi-governance coordination and synergies by the different administration levels (local, provincial and regional governments), multi-stakeholder collaboration between different sectors in industries and companies (value chains in priority sectors), consumption and production patterns, resource efficiency preventing the generation of waste and promoting the use of secondary raw materials. Moreover, they include the introduction of explicit references to the CE in regional strategies definition. In addition, they provide an overall framework from each region for a better overview on current initiatives and projects defined within the context of RETRACE.

RAPs made visible the benefits of adopting systemic approaches in the transition towards a CE and providing a pathway with necessary steps needed to undertake with that purpose, including policy recommendations for the update of regional/national Smart Specialization strategies to the transition to more systemic approaches. In addition, another component to ensure a successful learning process is the involvement of all stakeholders as end-users of the RAP and at the same time consider them as key players on disseminating this knowledge about CE and SD among their networks. [63]

*4.3. A systemic Assessment of RETRACE Policy Outcomes*

The RAPs defined a total of 22 policy actions to be implemented and monitored in Phase 2 of the project (Table 3). Each region has implemented different types of actions aimed mainly at a regional and local level of governance (also national in the case of small Member States such as Slovenia). The actions can be included in four main categories:

1.  **Policy strategies:** promoting multi-governance level initiatives addressing the ERDF (The European Regional Development Fund is a fund allocated by the European Union that aims to strengthen economic and social cohesion in the European Union by correcting imbalances between its regions ) or other regional policy instruments to direct regional strategies towards the CE and to increase stakeholder participation in circular businesses.

2.  **Call for proposals:** strengthening of CE-related topics and cross-sectoral scope of projects submitted in response to regional calls (mainly based on the ERDF instruments).

3.  **Pilot project implementation:** implementing pilot projects within key value chains for the regional CE, often in synergy with other funding instruments.

4.  **Training and education:** training activities aimed to different target groups (students, enterprises, public institutions) to create new knowledge and a cultural background favorable to the CE.

**Table 3.** Results of the actions implemented within the regional action plans.

| Action | Policy Gap Addressed | Barriers Addressed | Result | Success/Failure/Delay Factors |
|---|---|---|---|---|
| **Region 1 (Piedmont)** | | | | |
| Call for proposals addressed to regional enterprises | Policy in support to business and market development for CE activities/Support collaboration between sectors | Economic; technological | achieved | Success: Rely on internal funding scheme |
| Call for proposals addressed to regional stakeholders | Tailored policy measures on CE/Support collaboration between sectors/Policy in support to business and market development for CE activities | Economic; technological; institutional | achieved | Success: Rely on internal funding scheme |
| Training and education activities aimed at university students | Raising involvement and knowledge of operators concerning CE | Cultural; information | achieved | Success: Rely on internal resources/core business |
| Policy strategies to increase attention to CE | Raising involvement and knowledge of operators concerning CE/ Support collaboration between sectors/Policy in support to business and market development for CE activities/Tailored policy measures on CE | Economic; political; institutional | delayed | Delay: Longer time horizon |
| Policy strategies to support CE projects | Raising involvement and knowledge of operators concerning CE/Policy in support to business and market development for CE activities | Economic; regulatory | not achieved | Failure: Longer time horizon + higher organizational complexity |
| **Region 2 (Basque Country)** | | | | |
| Call for proposals addressed to regional enterprises | Policy focused on SMEs and micromanufacturing | Economic; information; technological | achieved | Success: Rely on internal resources/core business |
| Training and education activities aimed at professionals | Raising involvement and knowledge of operators concerning CE | Cultural; information | achieved | Success: Rely on internal resources/core business |
| Call for proposals addressed to regional stakeholders | Support collaboration between sectors | Economic; technological | achieved | Success: Effective stakeholder involvement |
| Implementation of a CE pilot project | Support collaboration between sectors | Economic; technological; political | achieved | Success: Effective stakeholder involvement |
| Policy strategies to support CE projects | Policy focused on SMEs and micromanufacturing | Economic; technological | partially achieved | Partial failure: Difficult in stakeholder/authority engagement |
| Implementation of a CE pilot project | Support collaboration between sectors | Economic; technological | not achieved | Failure: Rely on external funding sources/ external support |
| **Region 3 (Nouvelle Aquitaine)** | | | | |
| Training and education activities aimed at professionals | Support collaboration between sectors | Cultural; information | achieved | Success: Success in getting external funding resources + Effective stakeholder involvement |
| Call for proposals addressed to regional stakeholders | Tailored policy measures on CE | Economic; political; environmental | partially achieved | Partial failure: Difficult in stakeholder engagement + rely on external funding |
| Call for proposals addressed to regional stakeholders | Policy focused on SMEs and micromanufacturing | Economic; environmental | delayed | Delay: Difficult in stakeholder/authority engagement |
| Training and education activities aimed at professionals | Raising involvement and knowledge of operators concerning CE | Cultural; information | partially achieved | Partial failure: Longer time horizon |

**Table 3.** *Cont.*

| Action | Policy Gap Addressed | Barriers Addressed | Result | Success/Failure/Delay Factors |
|---|---|---|---|---|
| **Region 4 (Slovenia)** | | | | |
| Policy strategies to increase attention to CE | Policy regulations on the CE | Regulatory | not achieved | Failure: Difficult in stakeholder/authority engagement |
| Policy strategies to increase attention to CE | Policy in support to business and market development for CE activities | Economic; political | partially achieved | Partial failure: Longer time horizon + higher organizational complexity |
| Policy strategies to support CE projects | Policy focused on SMEs and micromanufacturing Tailored policy measures on CE | Economic; political; technological | achieved | Success: Effective stakeholder involvement |
| Training and education activities aimed at public institutions | Raising involvement and knowledge of operators concerning CE, | Cultural; institutional | partially achieved | Partial failure: Rely on external support |
| **Region 5 (Northeast Romania)** | | | | |
| Call for proposals addressed to regional enterprises | Support collaboration between sectors/Tailored policy measures on CE | Economic; political | achieved | Success: Rely on internal funding scheme |
| Training and education activities aimed at public institutions | Support collaboration between sectors/Raising involvement and knowledge of operators concerning CE /Policy regulations on CE | Cultural; institutional; environmental | not achieved | Failure: Rely on external funding sources/ external support |
| Implementation of a CE pilot project | Support collaboration between sectors/Policy regulations on CE/Policy in support to business and market development for CE activities | Economic; technological | not achieved | Failure: Rely on external funding sources/ external support |

The analysis of the results achieved or disregarded by RAP actions highlighted the main factors that determined the success or the total/partial failure in their implementation. Overall, there are no significant differences between different regions or different types of action, while the way the actions were implemented is relevant. Figure 5 shows that the actions successfully implemented: (i) leveraged existing policy instruments that were affected from the early stages of the project through direct involvement of their managing authorities (mainly ERDF); (ii) optimally exploited the skills and core businesses of the partners and stakeholders involved, especially with regard to training actions.

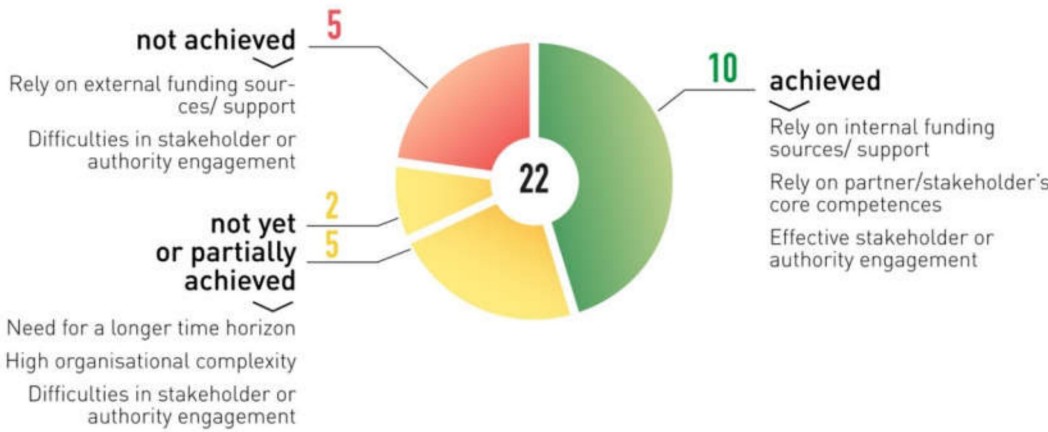

**Figure 5.** Regional action plan (RAP) implementation results and the related success/failure factors.

Conversely, it is worth pointing out that the actions that did not achieve the set objectives (i) relied on policy and funding instruments external to the project that were not reached within the set time frame (e.g., pilot project implementation); (ii) faced difficulties in building an effective engagement process of key stakeholders or regional authorities, in many cases this was due to the lack of common and shared objectives and effective ways of dialog and involvement. The actions not yet or partially implemented confirm the importance of building a stakeholder engagement process capable of tackling the organizational and bureaucratic complexity of the public sector, seeking to implement common projects that bring together different stakeholders and sectors.

The project results show how SD methods have been effective in identifying policy gaps and in defining actions tailored to the needs and tools available to each region. However, also the implementation process should adopt a systemic approach to stakeholder engagement and activity development, based not only on the potential of the region, but also of the specific partners and stakeholders involved in a policy action.

*4.4. Implementation of Systemic Change Over Time: Beyond RETRACE Time Frame*

RETRACE operated in the field of interregional policy-making, therefore the time-frame of a 4-year project is a short time to generate an immediate policy change as those processes usually take longer. In this perspective, it is permissible to conceive the implementation impacts to go beyond the end of the project. This process exposes the "actionable and proactive" mindset of the SD that combines with a foresight vision, can achieve concrete outcomes in the short term while addressing broader actions in the long term. From this point of view, SD ensures the introduction of policies were all participant actors can look after the development of the several actions on CE within a short, medium and long term, where the short and medium ones designed to support future implementations.

To illustrate how RETRACE implementation is framed on a long-term horizon effect on CE policy-planning process in EU regions, it is worth to narrow an insight of how the RAP actions can have an impact on the medium- and long-term horizon.

### 4.4.1. Short-Term Results

These results refer to the immediate policy outputs, which were delivered within the duration of the project. As RAP actions arrived on the current 2014–2020 Regional Operational Programme (detailed plan in which each EU region set out how money from the of the European Regional Development Fund will be spent during the programming period, specifies which of the 11 thematic objectives that guide cohesion policy in the 2014–2020 programming period will be addressed through the funding available under the operational programs), there is an awareness that they cannot modify the current policy status. In this perspective, the nature of these short-term actions was from funding opportunities for sectors with potential in CE, training workshops on CE, or open reviews of the regional strategies. These actions aimed to encourage managing authorities into scale up these results into significant impacts on a medium- and long-term policy perspective. Therefore, this approach across partners regions, including the involvement of all regional actors as end-users can anticipate future strategies on the mid—long term horizon, addressing them with today's changing drivers (e.g., research and innovation initiatives, technological development, active economic sectors with leading industries, key change-makers).

### 4.4.2. Medium-Term Results

The mid-term impacts of the RAP will be reflected after the project finalization on the upcoming 2021–2027 regional operational program. In this perspective, RETRACE RAP actions can deliver an essential input in outlining an impactful way to address the CE policy goals on the future programming period for EU funds, in a more systemic and territorial way. Moreover, the echo of these successful actions can also have an impact at the national level, creating a broader spectrum of calls that promote the assign of more funds for projects related to CE. From a governance point of view, the RAP supports regional strategies to foster more research and investment measures on CE. Lastly, this RETRACE results can influence how regions can oversee the development of CE in the next six years.

### 4.4.3. Long-Term Results

In the long-term, the RETRACE outcomes should be framed on the targets that the EU has established for sustainable development by 2030. In this view, must consider some EU policies towards a CE as circular economy Package, EU Bioeconomy Strategy, and the EU Plastics Strategy [20]. Regarding this high-level CE targets of the EU by 2030, the RAP contribution is a suitable milestone as they are addressing regions to reach significant environmental-economical-social benefits through the intervention of regional funds. All of the above resumes, it is possible to oversee the transformation of the RAP actions towards EU 2030 CE targets.

The case of RETRACE delivers an example of how short-term actions on wicked scenarios such as the linear economy framed in a holistic approach can deliver a more powerful impact on the long-term. This SD approaches on policy proofs that the transition to CE will require this kind of methodology as it considers the wicked problems that a transition to the CE can bring to European regions.

## 5. Discussion

The vast transformation the CE will carry in the following years inevitably will change the EU panorama, designing new geographies from an economical-social-environmental perspective, creating synergies between economic and environmental goals. From this point of view, the CE is amongst the most pressing challenges Europe has to face. More and more, actions are required at all levels of EU governance, as tangible results will be only possible through key change drivers as municipalities, and regions. It is indisputable that the advantages that a CE transition offers; however, it is necessary a systemic policy approach that will ensure a cohesive transition within EU regions and prevents the unknown economical-social effects this transition might bring.

In this view, this paper established an approach on how the SD methodology can deliver a policy design process within profound holistic comprehension of EU regions, allowing an appropriate diagnosis of the gaps and potential assets that can impact sustainable development.

The results of the desk research and field research highlight the complexity of designing policy actions to support stakeholders in overcoming the barriers to the CE. It is also evident the need for a systemic approach to tackle the complexity of circular socio-technical systems while avoiding the risk of simplification or excessive specialization of the proposed solutions. To this purpose, SD methods have demonstrated their effectiveness in the RETRACE research process, however, there are several constraints to be acknowledged, some of them have been already pointed out.

The holistic diagnosis based on the principles of co-production of knowledge in transdisciplinary research for multi-stakeholder process established reliable cooperation network with local stakeholders and managing authorities [67]. In this view, the holistic diagnosis to the development of the RAP, it required a further direct engagement with civil servants, industry, and community representatives which demonstrate that the SD approach increased awareness on CE local resources, opportunities, and challenges. However, the process acknowledged the strong presence of the cultural barrier rooted in the current cultural paradigm base on the traditional structures with linear thinking and a top-down approach. In the beginning, this entailed some difficulties in gathering the stakeholder workgroup. To overcome such barriers, the holistic diagnosis was a crucial component in the RAP decision-making processes, easing to visualize, from a broader lens within the collection of various feedbacks from the multi-stakeholder group. Moreover, SD also allowed visualizing the main barriers to wicked problems translated into the policy gaps.

The experience gained in the RETRACE project has proved how the stakeholder engagement is fundamental to build a shared and bottom-up policy design process. Specifically, the engagement process had success because the stakeholders through the SD were stimulated to think holistically and to overcome divisions between different sectors and roles, pursuing shared objectives. However, the results show that a proper engagement process in the policy-making phase does not guarantee the same engagement also in the policy implementation. Often the stakeholders involved in specific policy actions are different from the macro-actors that participated in defining it; moreover, it is difficult to involve new stakeholders and authorities after the policy has been designed. The stakeholder mapping phase is fundamental to create the group of actors to be involved in policy design, but this process should also be done when defining each policy action, so as to establish in advance who should be involved and how. This is even more essential when dealing with the public sector, where organizational complexity can jeopardize the success of a policy action.

The desk research highlighted how the CE requires thinking in longer time horizons and, therefore, enabling long-term decision-making. The four-year duration of the RETRACE project made it possible to implement actions in the medium and short term; however, partners deemed important to define some broader actions which, in fact, could not be finalized within the framework of the project. The main problem is not the failure to implement, but the lack of synergies with short-term actions that would ensure a gradual path beyond the project. Radical policy changes require a longer time frame, so it is important to design phased action plans that enable actions to be taken in the short term without losing sight of a long-term vision.

Finally, one of the key points emerged is the need to create synergies between different institutions to pursue joint policy actions on the CE. The RETRACE results show that most actions that sought support from external funding instruments or external institutions have failed. Without going into the specific cases, it is worth noting that the mediation role of the SD disciplines cannot focus only on stakeholders but must propose new methods to build inter-institutional collaborations.

## 6. Conclusions

Overall, cross-cutting projects like RETRACE demonstrate how systemic designers can have a leverage effect in the transition of the EU regions towards the CE. Exemplifying that such

dynamics are disruptive and transformative, as a practical methodology to enhance the CE transition. These results were supported by the RAPs, which brought a systemic policy approach for an effective CE policy-making, incorporating different policy interventions to boost the cooperation in Quadruple Helix [64]. In this way, creating a doorway where designers, managing authorities, and citizenship can effectively co-develop new policy opportunities. At the same time, activating the sphere of cooperation between governments and the design research community, the multi-stakeholder practice-led research becomes a mutual learning instrument to generate partnerships and policy development. These findings can become an inspiration model for policymakers in other regions that have to deal with common policy gaps in the transition process to a CE.

Hence, the RETRACE project outcomes can be approached in this concrete three innovative aspects for future practical applications on CE regional policymaking process;

1. *Projects focus on systemic and symbiotic approaches for a CE.* Introduce a much broader approach, focusing on design as a decision-making tool that supports sustainable development in the short, medium and long term. An SD project (RETRACE) represents a more elaborate version of the traditional CE project approach (cradle-to-cradle), building upon the broad approach of the project towards spatial planning (territory) and industrial perspectives, addressing a systemic approach at the regional level and the policy/program level. The outcomes of this examination show evidence that this topic should be further investigated at levels of national and global value chains.

2. *Decision-making instruments for regions and industrial value chains.* RETRACE project proof that a holistic diagnosis of regional framework conditions on natural assets and economic, productive and industrial links can effectively identify policy gaps and potential opportunities upon which to build more supportive policies. Furthermore, RETRACE delivered thus an assessment tool for regions, which in addition to the pool of knowledge and experience, and recommendations, facilitated the development of strategic thinking processes and the setup of systemic watch systems at EU regions. However, the research proof that regions are in the need of further instruments that allow them to perform a self-assessment procedure for CE strategies that more effectively supports coordinated policy-roadmaps within the EU regions.

3. *Focus on policy support and framework conditions.* RETRACE focused on policies and programs rather than on CE case studies conducted by specific companies or industries. The emphasis was placed on policies, programs, grants schemes, setup of observatories, etc., rather than on specific examples of waste reuse or valorization. These results prove that it is necessary to explore further the synergies between policy instruments and pilot actions to scale up CE innovation solutions.

Certainly, from the project emerged the dynamics of the most pressing issues that are hampering a transition to a CE at EU level. In this view, it is undeniable the policy roadmap that RETRACE has delivered on critical areas such as regulatory, economic, institutional and cultural, to achieve completely its purpose will require a long-term commitment from the regional, national and EU authorities.

**Author Contributions:** Conceptualization, C.G.N., A.P. and S.B.; Investigation, C.G.N. and A.P; Methodology, C.G.N. and A.P., and S.B.; Supervision, S.B.; Validation, S.B.; Visualization, C.G.N. and A.P Writing—original draft, C.G.N. and A.P.; Writing—review & editing, C.G.N. and A.P and S.B. All authors have read and agreed to the published version of the manuscript.

**Funding:** The RETRACE project was funded by Interreg Europe (2016–2020), grant number PGI00045.

**Acknowledgments:** A special acknowledgement to RETRACE partners: Politecnico di Torino, Regione Piemonte (IT), Azaro Foundation (ES), Beaz (ES), Higher School of Advanced Industrial Technology ESTIA (FR), Association for Environment and Safety in Aquitaine APESA (FR), Slovenian Government Office for Development and European Cohesion Policy (SI), Romanian North-East Regional Development Agency (RO), who support the process of this research project.

**Conflicts of Interest:** The authors declare no conflict of interest.

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
