# Peer review of "Systemic Design for Policy-Making: Towards the Next Circular Regions"

_sustainability, doi:10.3390/su12114494_

Round 1

Reviewer 1 Report

A timely article referring to the RETRACE project. Overall a valuable addition to the literature.

Minor things:

  • can you be more "tidy" in sorting the presentation of the theory and concepts please - your argumentation is somewhat difficult to follow
  • English grammar - some sentences need to be rephrased to proper English; examples are line 291, line 297, line 327 page 7, line 349 page 8, lines 697-699 page 18, and many more
  • social aspects like culture, agency, identity should be integrated more into designing CE systems, e.g. line 267 page 5
  • SD and HD are explained, then the different policy results of the regions are presented, but then the discussion should include more how SD and HD help(ed) to tackle such challenges
  • the literature of transdisciplinarity (TD) is not sufficiently discussed here - continuous joint problem framing is what relates to the "engagement process in the policy-making phase) line 676 page 19. this would have to be added to the discussion.

Author Response

Dear Reviewer,

Thanks for your comments to our work, we add most of your comments.

can you be more "tidy" in sorting the presentation of the theory and concepts please - your argumentation is somewhat difficult to follow.

  • As a general recommendation from most of the reviewers, the introduction was reframed and synthesized.

social aspects like culture, agency, identity should be integrated more into designing CE systems, e.g. line 267 page 5

  • In the part noted by the reviewer, the methodology is stated in general terms, not targeting circular economy per se. However, this approach was specifically targeted in Table 2, where is described step by step how the Systemic Design addresses

English grammar - some sentences need to be rephrased to proper English; examples are line 291, line 297, line 327 page 7, line 349 page 8, lines 697-699 page 18, and many more

  • Grammar was reviewed on the indicated places and further corrections where made.

SD and HD are explained, then the different policy results of the regions are presented, but then the discussion should include more how SD and HD help(ed) to tackle such challenges

  • As a request of the other reviewers in the conclusions and discussion was included a concrete set of recommendations that the project produced and that should be applied in future projects and research.

the literature of transdisciplinarity (TD) is not sufficiently discussed here - continuous joint problem framing is what relates to the "engagement process in the policy-making phase) line 676 page 19. this would have to be added to the discussion.

  • References to transdisciplinary research were added, as well as its influence on co-design instruments such as the Holistic diagnosis.

Hope the new draft has improved.

Best Regards

Carolina Giraldo Nohra

Reviewer 2 Report

Dear authors,

The paper presents a methodology applied to facilitate the implementation of circular economy at regionallevel. It is a result of an interreg project.

I would like to thank the authors for this complete paper, that not only presents a method but also tests it with local actors and discuss relevant observations. It is useful to overcome the barriers for the implementation of CE.

I recommand the paper for publication

Best regards

Author Response

Dear reviewer,

Thanks for your comments and compliments, we are very satisfied with the results of this research.

Best Regards

Carolina Giraldo Nohra

Reviewer 3 Report

The manuscript is well written and organized to explore the systemic design for policy-making. Some comments are shown as below.

  1. The capital on the manuscript title should be used. “Systemic Design for Policy-Making: Towards the Next Circular Regions”
  2. Line 51, “…by Korhonen, Honkasalo, Seppala (2018) [18] brings…” should be “…by Korhonen et al. [18] brought…”
  3. Line 209, “…by Buchanan (1992) [37] who focus on…” should be “…by Buchanan [37] who focused on…”
  4. Line 359, “However, Williams (2019) [62] and Vermunt (2019) [63] deliver…” should be “However, Williams [62] and Vermunt [63] delivered…”
  5. Line 405, the subsection title should be “4.1 Stakeholder Engagement Processes”.
  6. Line 441, the subsection title should be “4.2 Systemic Design Research Methodology for Circular Policies”.
  7. Line 476, please delete the word “(Sevaldson, 2011)”.
  8. Lines 503-504, what is the “(Author, 2018)”?
  9. Line 554, the subsection title should be “4.3 A Systemic Assessment of RETRACE Policy Outcomes”.

10) Line 595, the subsection title should be “4.4 Implementation of Systemic Change over Time beyond RETRACE Time Frame”

11) The Section “Discussion and conclusions” should be separated into two sections “Discussion” and “Conclusions”.

Author Response

Dear Reviewer,

Thanks for your comments and input  to our work,

All of the grammar and editing remarks you underlined were addressed. Furthermore, the discussion and conclusion section was divided on your request.

Hope the new draft has improved.

Best Regards

Carolina Giraldo Nohra

Reviewer 4 Report

The article studies how the System Design methodology can deliver a Policy Design process within profound holistic comprehension of EU regions, allowing an appropriate diagnosis of the gaps and potential assets that can impact sustainable development. The article is strictly theoretical, that is why the presented theses are hard to prove. However, in my opinion the presented theses should be subject to scientific discussion, and the best way is to publish them. I have two minor remarks:

1) Introduction section is very extensive. Therefore, the clear aim of the research is hard to find in this section. The aim should be precisely defined and explained. in the separated paragraph. Moreover, a scientific contribution of the article should be listed.

2) Conclusions section should be supplemented with directions of future research and potential practical applications of the research.

Author Response

Dear Reviewer 

Thanks so much for your comments and inputs to our work.

On regards your remarks:

1) Introduction section is very extensive. Therefore, the clear aim of the research is hard to find in this section. The aim should be precisely defined and explained. in the separated paragraph. Moreover, a scientific contribution of the article should be listed.   

We have shortened the introduction section to make it more focused on the two topics of the article (policy-making for circular economy and Systemic Design). We have also added a paragraph on the research questions to better explain the specific objectives of the research, highlighting the scientific contribution of the article.  

  2) Conclusions section should be supplemented with directions of future research and potential practical applications of the research.   

We included an additional section on conclusions where we specified the concrete takeout of the project for future research or applications in other regions. We appreciate very much your contribution to our work  

Best Regards,   

Carolina Giraldo Nohra

Round 2

Reviewer 3 Report

I recommend that the revised manuscript can be accepted for publication.